# The comprehensive connectome of a neural substrate for 'ON' motion detection in *Drosophila*

Shin-ya Takemura[1]*, Aljoscha Nern[1], Dmitri B Chklovskii[2,3], Louis K Scheffer[1], Gerald M Rubin[1], Ian A Meinertzhagen[1,4]

[1]Janelia Research Campus, Howard Hughes Medical Institute, Ashburn, United States; [2]Simons Center for Data Analysis, Simons Foundation, New York, United States; [3]Neuroscience Institute, NYU Medical Center, New York, United States; [4]Department of Psychology and Neuroscience, Dalhousie University, Halifax, Canada

**Abstract** Analysing computations in neural circuits often uses simplified models because the actual neuronal implementation is not known. For example, a problem in vision, how the eye detects image motion, has long been analysed using Hassenstein-Reichardt (HR) detector or Barlow-Levick (BL) models. These both simulate motion detection well, but the exact neuronal circuits undertaking these tasks remain elusive. We reconstructed a comprehensive connectome of the circuits of *Drosophila*'s motion-sensing T4 cells using a novel EM technique. We uncover complex T4 inputs and reveal that putative excitatory inputs cluster at T4's dendrite shafts, while inhibitory inputs localize to the bases. Consistent with our previous study, we reveal that Mi1 and Tm3 cells provide most synaptic contacts onto T4. We are, however, unable to reproduce the spatial offset between these cells reported previously. Our comprehensive connectome reveals complex circuits that include candidate anatomical substrates for both HR and BL types of motion detectors.

*For correspondence: takemuras@janelia.hhmi.org

Competing interests: The authors declare that no competing interests exist.

## Introduction

A classic problem in visual neuroscience, how the visual system detects image motion, has recently undergone resurgent interest in the fruit fly *Drosophila*. The problem has long been analysed using mathematical models that compare two adjacent viewpoints, of which the Hassenstein-Reichardt (HR) elementary motion detector (EMD; *Hassenstein and Reichardt, 1956*; *Figure 1A*) or Barlow-Levick (BL) model (*Barlow and Levick, 1965*; *Figure 1B*) are both widely canvassed. Both models compare two visual inputs separated in time and space and simulate motion detection well (*Haag et al., 2016*). However, the exact neural mechanism underlying the detection of direction by motion-sensing cells is still understood only imperfectly.

Visual processing in flies occurs consecutively in four retinotopically organised neuropile relays beneath the photoreceptor layer (*Figure 1C*). Each neuropile comprises an array of repeating modules corresponding to the hexagonal lattice of ommatidia in the overlying compound eye (*Braitenberg, 1970*; *Fischbach and Dittrich, 1989*; *Takemura et al., 2008*). Visual signals generated by photoreceptors, R1-R6, are transmitted to two matched classes of large paired interneurons, L1 and L2, and a third, L3, in the first neuropile, or lamina (*Figure 1C*) (*Meinertzhagen and O'Neil, 1991*; *Rivera-Alba et al., 2011*). Like ON and OFF bipolar cells in the vertebrate retina (*Wässle, 2004*; *Dowling, 2012*), the visual pathways split at the first synapse, with L1 and L2 as the main inputs

(*Rister et al., 2007*), to separate ON- and OFF-edge motion pathways, respectively (*Joesch et al., 2010*; *Clark et al., 2011*).

ON-edge signals are then locally computed on the dendritic arbours of columnar cells, T4, in the most proximal stratum M10 of the second neuropile, or medulla (*Fisher et al., 2015*). These cells respond to visual motion in a directionally-selective manner (*Maisak et al., 2013*), and have four sub-types, T4a-T4d (*Fischbach and Dittrich, 1989*; *Takemura et al., 2013*). Each subtype is tuned to one of four cardinal directions of stimulus motion, front-to-back (T4a), back-to-front (T4b), upward (T4c) and downward (T4d) (*Maisak et al., 2013*), approximately one each per unit column of the medulla (*Takemura et al., 2013*; *Mauss et al., 2014*). Each in turn projects its terminal to a specific stratum, one of four in the motion-sensing fourth neuropile, the lobula plate (*Figure 1C*). There, T4 signals from hundreds of columns sum to generate wide-field motion responses in target, direction-ally-selective tangential cells (*Borst et al., 2010*), aided by various lobula plate interneurons (*Mauss et al., 2015*).

How do T4 cells compute the direction of motion? A recent connectomic study using serial-sec-tion EM (ssEM) identified two medulla cell types, Mi1 and Tm3, as the direct pathways between L1 and T4 (*Takemura et al., 2013*). Detailed analyses revealed that the weighted centre of the Mi1-mediated receptive field is displaced from that of the Tm3-mediated subfield, and that the direction of displacement between the two subfields aligns in ~70% of cases with T4's preferred direction (*Takemura et al., 2013*). Electrophysiological recordings demonstrate a temporal offset between the responses of the two cell types: Mi1 exhibits a peak light response later than that in Tm3 (*Behnia et al., 2014*), although the time difference (~15 ms) is probably too small to account for T4's measured temporal frequency optimum of ~1 Hz (*Maisak et al., 2013*). The direction of spatial off-sets from Tm3 to Mi1, generally consistent with T4's preferred direction, matches the requirements of non-delay and delay input channels of a BL-type detector.

Despite this evidence, there are multiple indications that Mi1 and Tm3 inputs are not the only players in T4's motion detection. First, a BL-type detector requires Mi1 and Tm3 inputs to T4 to have different signs (*Figure 1B*); this is not supported by available neurotransmitter information (see below). Second, whereas each T4 dendrite is orientated primarily in a direction aligned with its direc-tional preference (*Takemura et al., 2013*), Mi1 and Tm3 synapses on those dendrites are distributed with substantial overlap. Third, although recent experiments confirm a general role for Mi1 in ON-motion detection, they suggest that Tm3 is required for responses only to fast-motion stimuli (*Ammer et al., 2015*), so that other inputs must be active under slow-motion stimulus conditions.

Here, we use a superior EM technique to reconstruct T4 cells and their presynaptic partners, and provide a comprehensive anatomical view of this critical step in fly motion vision. Our results suggest new candidate circuits for motion detection, including previously unidentified inhibitory pathways.

## Results and discussion

### Medulla connectome reconstruction using Focussed Ion-Beam milling Scanning Electron Microscopy (FIBSEM)

To reconstruct T4's input circuits comprehensively we used a new method of focussed ion-beam mill-ing scanning electron microscopy (FIBSEM), which enabled a more comprehensive dense connec-tome with greater accuracy (see Methods). Briefly, we imaged a ~40 × 40 x 80 μm volume at an isotropic resolution of 10 nm per voxel, annotated ~53,500 presynaptic sites and ~315,500 postsyn-aptic sites, and reconstructed >900 neuron bodies including >60 T4 cells. The dataset volume com-prised seven medulla columns: a central 'Home' column and its six immediate neighbours (*Figure 1D*). These seven columns have been partially analysed elsewhere (*Takemura et al., 2015*); here we focus on the inputs to T4.

### Synaptic inputs to T4 cells from eight cell types

Since a T4's dendrites spread across approximately seven columns, the dendrites of the T4 cells in the Home column should be completely contained in the reconstructed volume. We found Mi1-Home contacted >20 T4 cells, forming 2–43 synapses onto each. Among these, only four T4s received >35 such contacts. They included one of each subtype, T4a-T4d, as identified by the dis-tinctive orientation of their dendrites (*Takemura et al., 2013*) (*Figures 2A* and *4C–4E*).

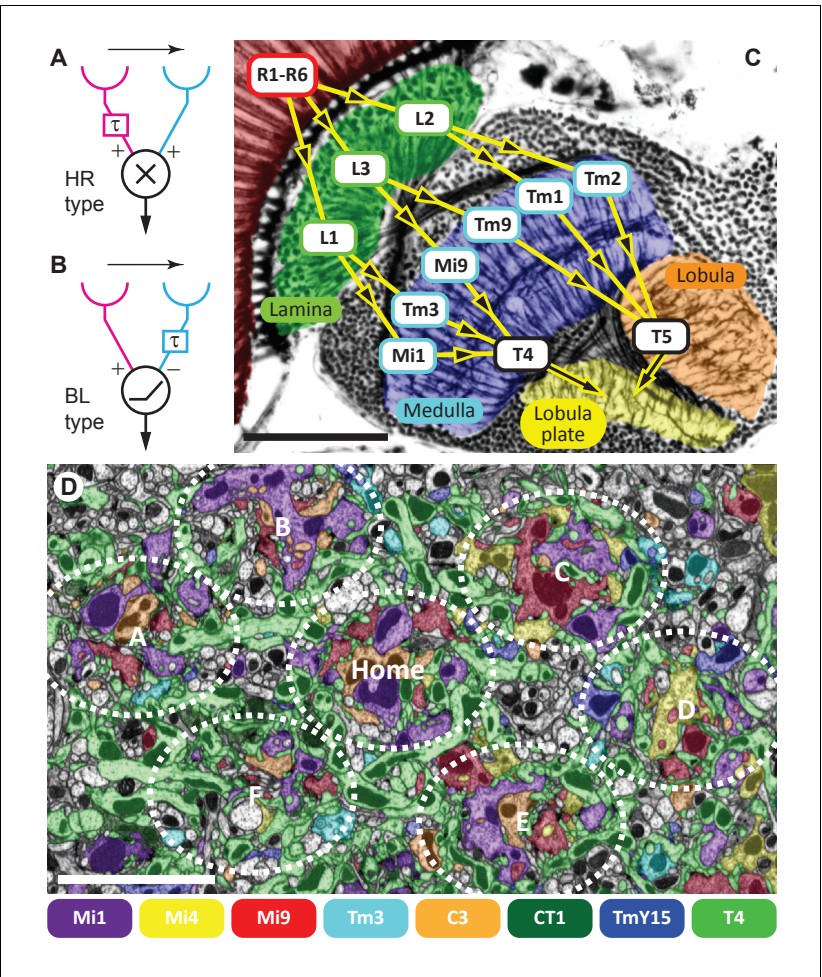

**Figure 1.** Visual motion-detection pathways and circuits in *Drosophila*. (**A, B**) Diagrams of two major predicted models of motion detection: (**A**) the Hassenstein-Reichardt (HR) and (**B**) Barlow-Levick (BL) type. In both models, two input channels with spatially offset fields of view, one with a delay component (τ), are combined non-linearly to generate directionally-selective responses. The illustrated detectors respond preferentially to rightwards-moving objects. (**C**) Bodian-stained horizontal section of the *Drosophila* optic lobe revealing the four optic neuropiles, lamina, medulla, lobula and lobula plate. Visual signals generated by photoreceptors R1-R6 are encoded by second-order interneurons in the lamina. ON-edge signals are fed by L1 and transmitted to direction-selective T4 cells via medulla interneurons (e.g. Mi1 and Tm3). OFF-edge signals are fed by L2 to direction-selective T5 cells in the lobula via transmedulla neurons (e.g. Tm1 and Tm2). (**D**) EM cross section at the level of T4 dendrites in medulla stratum M10. Neurites of T4 cells and their input neurons are colour-labelled (key). Unlabelled profiles are mostly axons that penetrate this layer without synapsing onto T4s. Seven reconstructed columns (Home, and A through F) are roughly outlined (dotted ovals). Scale bars, 50 µm (**A**) and 5 µm (**D**).

Next, we sought other cell types with input to the T4s. Reconstructed cell shapes revealed eight medulla neuron types that account for inputs to the T4 dendrites (*Figure 2A*). We confirmed previously reported inputs from neurons Mi1, Tm3, C3, Mi4, Mi9 and T4 itself (*Takemura et al., 2013*). Comparing the remaining reconstructed cells with light microscopy data further identified a previously undescribed TmY neuron, which we call TmY15 (*Figure 2A and B*; *Figure 2—figure supplement 1*), and neurites of a new large tangential cell, which we call CT1 (*Figure 2A and C–E*; *Figure 2—figure supplement 1*). CT1 spans the opposing faces of medulla stratum M10 and lobula stratum Lo1 that contains dendrites of T4's counterpart, T5, as a single giant neuron (*Figure 2C*). Its terminal arbours show a strikingly regular pattern in each medulla and lobula column (*Figure 2D*;

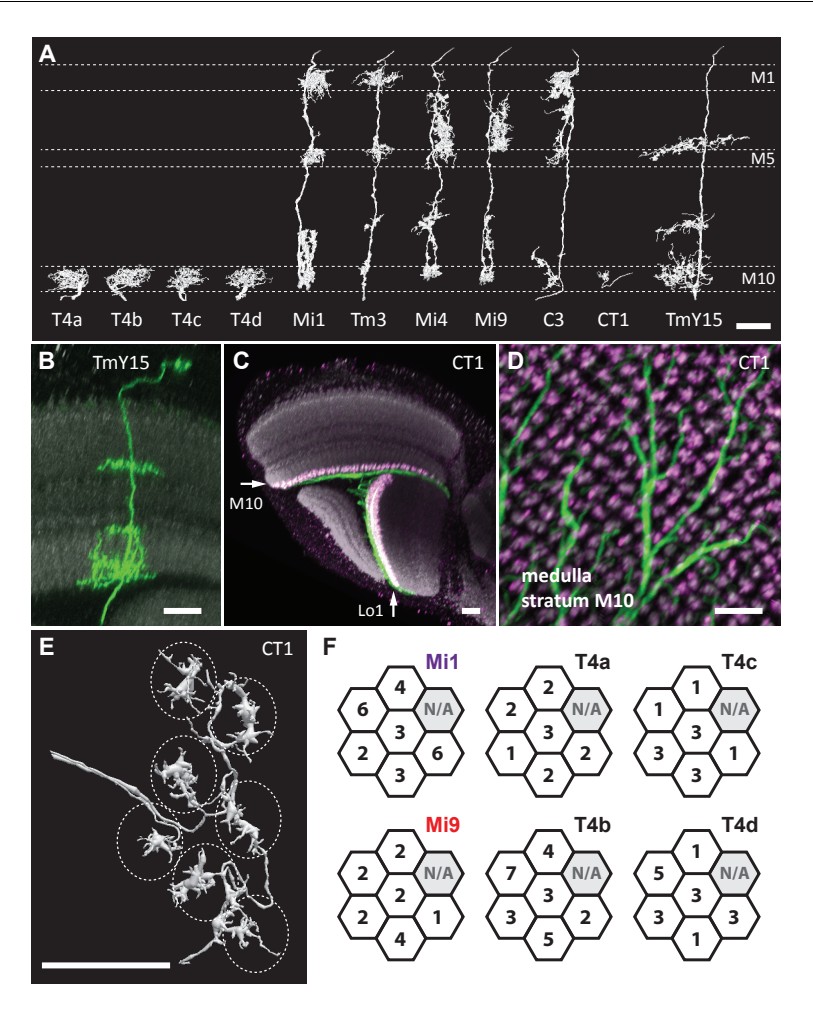

**Figure 2.** Eight cell types that provide >96% of synaptic inputs to T4 cells. (**A**) Reconstructions of four T4 subtypes (T4a-T4d) identified in the central 'Home' column, and seven other cell types providing synaptic input to these. T4 also connects reciprocally with other T4s and is hence considered the eighth type. (**B**) Light microscopy of single-cell labelling for the medulla arbours of a TmY15 cell. TmY15 arbours extend to >10 medulla columns in stratum M10 (see *Figure 2—figure supplement 1* for additional TmY15 anatomy). Grey: a neuropile marker. (**C**) Light microscopy of a single CT1 cell widely innervating M10 and distalmost lobula stratum Lo1. The cell body is located in the central brain (see *Figure 2—figure supplement 1*). Green: membrane marker; magenta: a presynaptic marker, both expressed in CT1; grey: a general neuropile marker (For details see Experimental Procedures). (**D**) Plan view of the CT1 cell in (**C**) generated from the same confocal stack. In medulla stratum M10, CT1 sends its arbour terminals to each medulla column, tiled highly regularly. (**E**) EM-reconstructions of CT1 terminal arbours in M10. CT1 terminals present in each column match light microscopy images. (**F**) CT1 is postsynaptic to Mi1, Mi9 and T4 cells. Synaptic inputs from these cell types shown separately for the CT1 terminal in each of the seven columns. In each column, CT1 receives a few (1-7) inputs from all cell types. The arbour in the upper-right column (N/A) was not reconstructed because it lay too close to the volume edge. Scale bars, 10 μm throughout.

The following figure supplement is available for figure 2:

**Figure supplement 1.** Additional TmY15 and CT1 anatomy.

---

*Figure 2—figure supplement 1*), which is also confirmed by EM reconstructions (*Figure 2E*). In each column, CT1 receives a few synaptic inputs from Mi1, Mi9 and all T4 subtypes (*Figure 2F*).

We next examined the number of T4 inputs from each of these cell types (*Figure 3*). Each T4 receives >200 synaptic inputs in total, and Mi1 and Tm3 cells are indeed the largest contributors

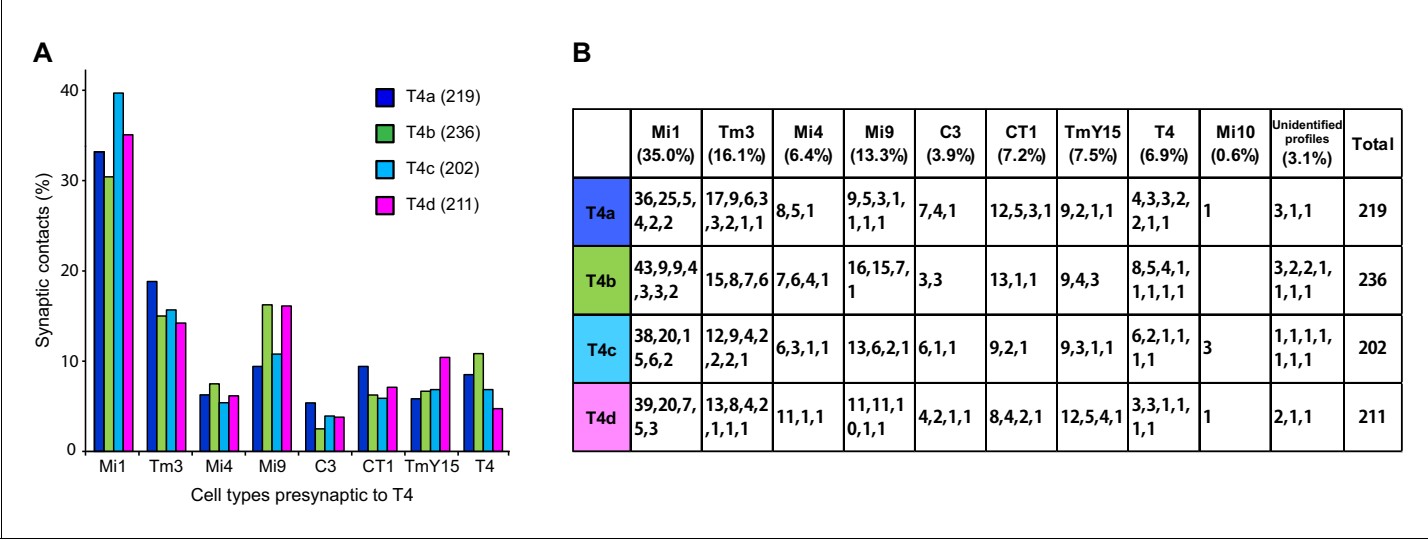

**Figure 3.** Total counts of synaptic inputs to T4 subtypes of the Home column. (**A**) Fraction of the total synaptic connections for eight cell types presynaptic to T4s. Numbers in parentheses (key, upper right) are the gross synaptic input counts to each T4 subtype. (**B**) Number of synaptic inputs from individual presynaptic cells to the four T4 subtypes. Percentages for each input cell type are the mean fractions among the four T4 subtypes. Note that connections having only a single synapse are not used in the plots in *Figure 4A*.

with 35.0% and 16.1% of the total, respectively (*Figure 3*). Endorsing improved completeness of the new FIBSEM reconstruction, total synaptic inputs from Mi1 and Tm3 to the four T4 subtypes (116, 109, 113, and 104, respectively), totalled 33% more than in the previous ssEM connectome (97, 69, 85 and 81) (*Takemura et al., 2013*).

Additional unicolumnar cell types, Mi4, Mi9 and C3, all provide fewer synaptic inputs to the T4s than Mi1 and Tm3. However, we identified more than previously reported, primarily because many of these synapses are located in the periphery of a T4's dendrites and were therefore excluded from an earlier account (*Takemura et al., 2013*), which sampled densely only a single column. Secondarily, we benefit from greater image quality of FIBSEM. Their wider contribution is nevertheless significant: 13.3% (Mi9), 6.4% (Mi4) and 3.9% (C3), in aggregate ~24% of the total (*Figure 3*). The new FIBSEM reconstruction identified many more connections than the previous ssEM study which found 4 Mi4 inputs, 6 Mi9 inputs, and 5 C3 inputs (*Takemura et al., 2013*).

The newly identified TmY15 and CT1 cells together account for another ~15% of T4 inputs (*Figure 3*). In addition, a T4 cell also connects reciprocally to other T4s in neighbouring columns. Only very few synapses onto T4s were from other cell types or could not be identified: an Mi10 neuron makes a few input synapses (~0.6%), with the remaining 3.1% from several unidentified profiles that left the reconstructed volume, each contacting a T4 at ≤3 synapses (*Figure 3B*). At least some of these unidentified synapses are likely to be from branches of other T4s that reside outside the imaged region. We therefore reason that our dense FIBSEM reconstruction captures all input cell types to T4 and is now reliably definitive.

## Spatial offset between different inputs distributed over T4 dendrites

All models of motion detection require comparison between inputs that view different points in visual space. To judge the potential relevance of T4's input neurons to generating motion direction preference, we analysed the distribution of synapses over the T4 dendrites (*Figure 4C–E*) and, for cell types that could be unambiguously assigned to single columns, across visual columns (*Figure 4A*).

Each T4 subtype receives input from multiple Mi1s, with the strongest from the Home Mi1 and weaker inputs from surrounding Mi1s (*Figure 4A*); these spread along the shafts of the T4 dendrites (*Figure 4C*). Likewise, Tm3 synapses appear to distribute evenly over the T4 dendrites (*Figure 4D*). However, while Mi1 cells can be readily assigned to specific columns, Tm3s occupy more variable

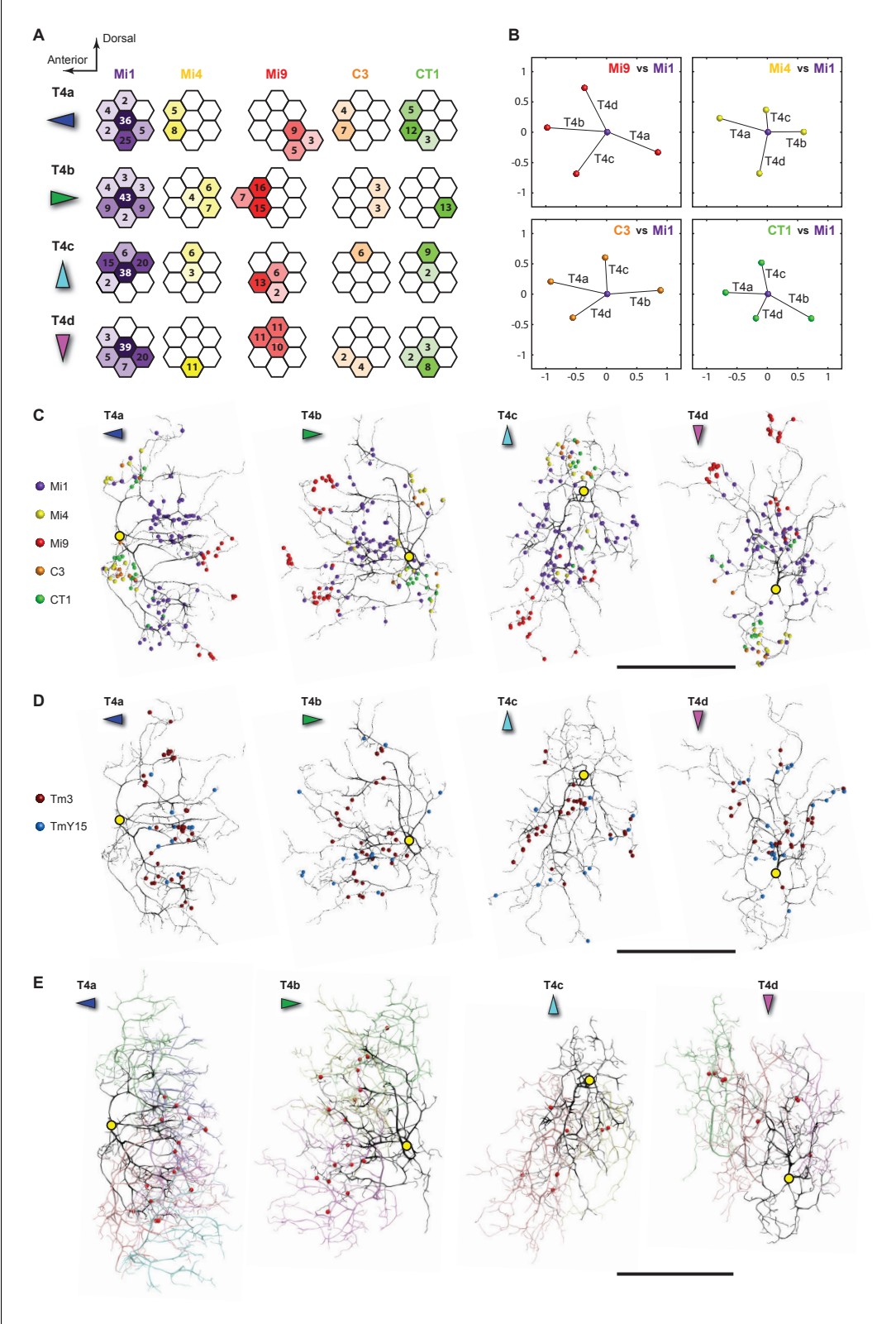

**Figure 4.** Distributions of synaptic inputs onto T4. (A) Counts of synaptic inputs to T4s from each class of input neuron from neighbouring columns are shown in the corresponding hexagonal array. Cell types for which individual neurons cannot be unambiguously assigned to single columns are not included. Thus, Tm3 cells cannot be assigned to a particular column because they are multicolumnar (*Figure 4—figure supplement 1*). Analysis on another set of T4s confirms the results of the Home column T4s (*Figure 4—figure supplement 3*). (B) Angular subtense between weighted anatomical

*Figure 4 continued on next page*

*Figure 4 continued*

subfield centres for Mi1 paired with four other medulla cell input neurons for the four T4 subtypes. Axes of X and Y show centre-to-centre distances between adjacent columns. The plots show considerable spatial displacements, in some cases more than an inter-ommatidial distance (see *Figure 4— figure supplement 2* for the limited analyses of Tm3 to Mi1-offset). (C) Distributions of synaptic inputs onto T4 dendrites. Colours of puncta correspond to presynaptic cell types. Yellow circles indicate the locations of the axon's main trunk, relative to which dendrites spread in one of four predominant directions. The locations of inputs from Mi9 (red) are spatially segregated from those of Mi4/C3/CT1 (yellow/orange/green). Mi1's inputs (purple) spread between these locations. These data are used to compute the anatomical subfield centres in (B). (D) Distributions of synaptic inputs from Tm3 and TmY15. Synapses are rather uniformly distributed over the T4 dendrites. (E) Distributions of synaptic inputs from T4s in the surrounding columns. The T4 subtypes in the Home column (black) each receive input from other T4s (colours) that have the same dendritic branch orientation (i.e. the same preferred direction). Red puncta indicate the synaptic contacts from the neighbouring T4s onto the T4s. Scale bar, 10 µm (C–E).

The following figure supplements are available for figure 4:

**Figure supplement 1.** Positions of Mi1, TmY15 and Tm3 arbours relative to medulla columns in stratum M10.

**Figure supplement 2.** Limitations of the reconstructed volume and analyses.

**Figure supplement 3.** Synaptic inputs to T4 cells in Column E.

positions in the column periphery, and cannot be so assigned (*Figure 4—figure supplement 1*). Calculating anatomical receptive fields (*Takemura et al., 2013*), is also limited because wide collaterals of Tm3 cells in strata M1 and M5 (*Figure 2A*) extend beyond the 7-column region. The reconstructed Tm3s therefore receive some input from columns not contained in the FIBSEM image stack used here and which we therefore could not trace. This limits the accuracy with which potential spatial offsets between Tm3 and other cell types could be determined (see *Figure 4—figure supplement 2A* and Discussion).

Inputs to T4 from Mi4 and Mi9 are noticeably segregated from each other (*Figure 4A and C*). Significantly, Mi9's input synapses always cluster at the tips of the T4 dendrites while Mi4's inputs localize to their bases (*Figure 4C*). Inputs from other T4 cells also clustered towards T4's dendrite tips (*Figure 4E*). The T4-T4 synapses showed striking subtype selectivity connecting with cells with the same direction preference, apparently qualifying a group of T4s to signal each other about a particular stimulus direction. This subtype selectivity was observed without exception in every T4-T4 contact. The retinotopic positions of the presynaptic T4s lie relative to the postsynaptic T4 in a direction opposite to the cell's preferred direction (*Figure 4E*).

Like Mi4, C3 and CT1 also made synapses at the bases (*Figure 4C*). Even though CT1 is a widefield neuron (*Figure 2C*; *Figure 2—figure supplement 1*), its connections are displaced from the Mi1 centre in a manner very similar to that of Mi4 and C3 (*Figure 4B*), suggesting that each columnar arbour may serve a local computational role. We therefore counted each column's CT1 arbour as a separate unit per column (*Figure 4A*). Finally, TmY15 is presynaptic to T4s with a branching arbour extending more widely than the 7-column area in stratum M10 (*Figure 2B*; *Figure 2—figure supplement 1*). We have reconstructed three TmY15 cells in the 7-column volume, of which two are only partial, suggesting that this cell type is distributed less frequently than one cell per column. Its synaptic inputs were rather uniformly distributed over T4's dendrites (*Figure 4D*).

To confirm the stereotypy of the different columnar distributions of T4 inputs, we extensively traced another set of T4s in a neighbouring column; again using inputs from Mi1 and dendritic morphology to identify four T4 subtypes associated with this column. These analyses supported the results for the Home column T4 cells (*Figure 4—figure supplement 3*).

## Polarity of transmission from the synaptic inputs onto T4

To gain insight into the possible signs of T4's inputs, we next sought information on their neurotransmitters. Immunolabelling studies have suggested that C3 is GABAergic (*Kolodziejczyk et al., 2008*) while Mi1 expresses choline acetyltransferase (*Hasegawa et al., 2011*; *Pankova and Borst, 2017*), suggesting it is cholinergic. Our immunolabeling results using anti-GAD1, anti-ChAT and anti-VGlut confirm these results for C3 and Mi1 and further suggest that GABA, acetylcholine and glutamate are the respective transmitters for Mi4, Tm3 and Mi9 (*Figure 5*). These results are also supported by

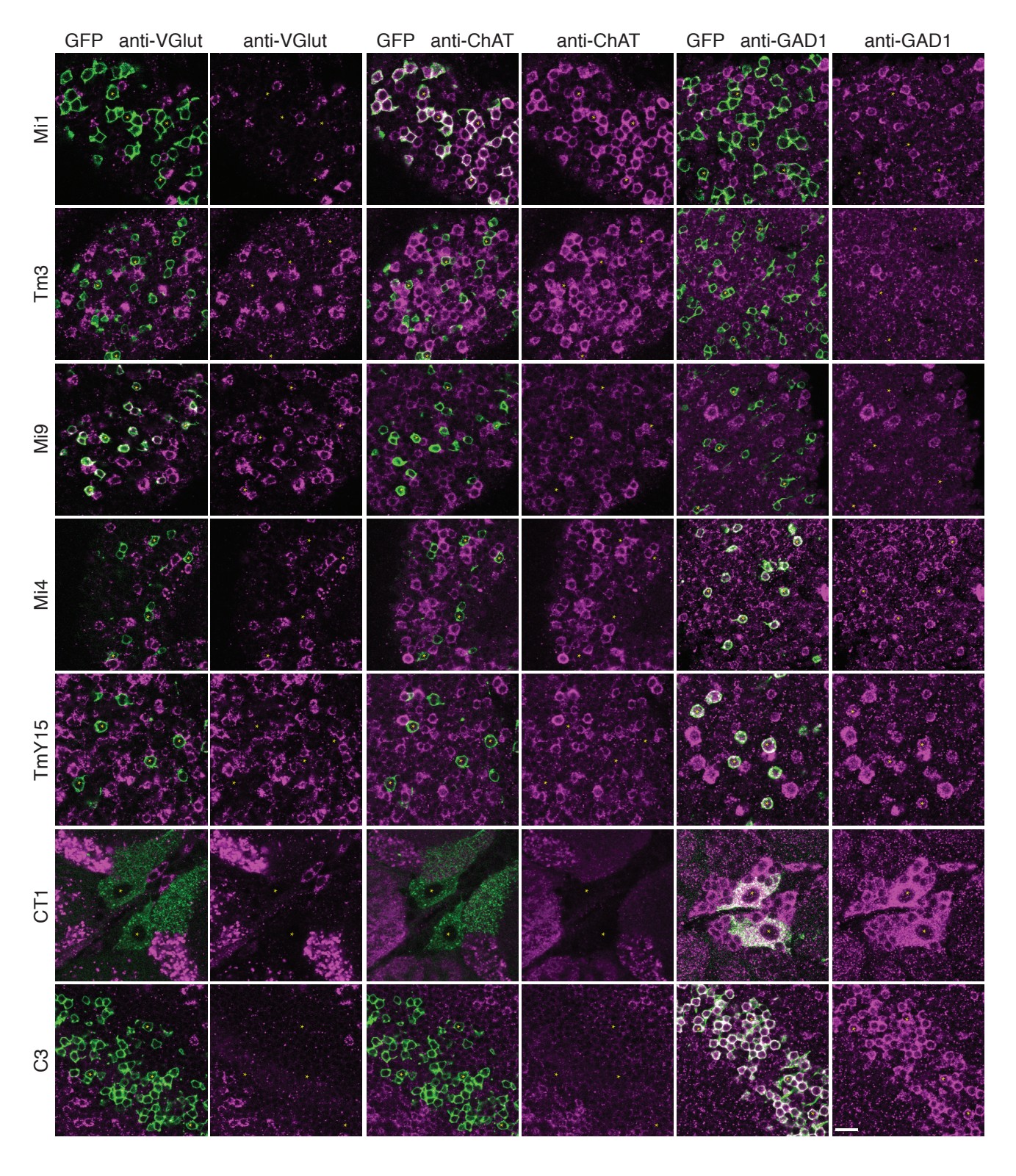

**Figure 5.** Anti-VGlut, anti-ChAT and anti-GAD1 immunoreactivity of T4's input neurons. Confocal sections showing GFP-labeled cell bodies. Small yellow asterisks are intended to facilitate comparison between the single channel panels. At least four brains (CT1) or at least two optic lobes from two brains (all other cell types) were examined for each condition. The main observed variation within a genotype was that the VT048653 driver line used as marker for TmY15, also included a small number of GAD1 negative cells (~12%; 15/130 cells counted) and VGlut positive cells (not counted). Supporting

*Figure 5 continued on next page*

*Figure 5 continued*

other recent evidence (*Pankova and Borst, 2017*), ChAT-immunoreactivity of Tm3 cells was detectable, although, under the conditions used here much weaker than that of Mi1 neurons. The GFP labeled arbours located to the top and right of the CT1 cell bodies in the VGlut and ChAT panels belong top a different cell type. ChAT expression in M1 and Tm3, GAD1 in Mi4 and C3 and VGlut in Mi9 were also observed in a transcriptomics study (L. Henry, F. Davis and S. Eddy, personal communication). Scale bar represents 10 µm.

cell-type-specific transcript profiling studies (L Henry, F Davis and S Eddy, personal communication). In addition, anti-GAD1 immunolabelling reveals that CT1 and TmY15 appear to use GABA (*Figure 5*).

These considerations indicate that T4 dendrites receive both excitatory and inhibitory inputs and, together with the EM data, allowed us to examine the distribution of inputs of different predicted signs: the synapses from C3/Mi4/CT1 onto the T4 dendrite base all appear to be GABA-ergic and therefore inhibitory, whereas those from Mi1 and Tm3 along the shaft of the T4 dendrite are putatively cholinergic, and therefore most likely to be excitatory (*Figure 6B*). Mi9's input at the T4 dendrite tip, putatively glutamatergic, may be either excitatory or inhibitory, depending on receptor expression at this synapse, given that glutamatergic neurotransmission can be inhibitory in *Drosophila* (*Liu and Wilson, 2013*; *Mauss et al., 2015*). T4-T4 synapses, also at the dendritic tips, are primarily cholinergic (*Mauss et al., 2014*; *Shinomiya et al., 2014*), and thus most likely to be excitatory.

## Candidate neuronal substrate for motion detectors

The finding of additional T4 inputs and their synaptic arrangements allows us to compare the predictions from functional studies with the requirements of different models of motion detection. Our findings reveal potential substrates for both BL- and HR-type motion detectors. A recent imaging study reported that T4 cells indeed use both HR model preferred-direction enhancement and BL model null-direction suppression (*Haag et al., 2016*). Which individual inputs onto T4's dendrites fulfill the requirements for enhancement and suppression of directional selectivity? Of the synaptic inputs to T4 that are anatomically qualified to serve as two channels of a motion detector, several could support BL motion detectors (*Figure 1B*). First, inputs from Mi4, C3 and CT1 onto T4's dendrite bases, which could form one input channel, are putatively inhibitory and therefore opposite in sign to excitatory input from Mi1 and Tm3 which could form another channel. Second, the vectors of physiological direction preference of each T4, as inferred from dendrite orientation, match those measured in a direction from excitatory inputs (dendrite shaft, Mi1/Tm3) to inhibitory inputs (dendrite base, Mi4/C3/CT1) in accordance with a BL motion detector (*Figure 6B*; cf. *Figure 1B*). Third, the inhibitory inputs arrive via a di- or trisynaptic pathway from L1, via L5 or Mi1 and others (e.g. L1→L5→Mi4→T4) (*Figure 6A*; see also *Figure 3a* in ref 16). By contrast, putatively excitatory inputs from Mi1 and Tm3 onto T4's dendrite shafts are monosynaptic (*Figure 6*) suggesting a possible time delay between excitatory and inhibitory channels, but one that would be too short alone to account for the temporal frequency optimum in T4 cells of about 1 Hz (*Maisak et al., 2013*).

Among the potential inputs that could serve as HR motion detectors we first consider Mi1 and Tm3, as previously suggested (*Takemura et al., 2013*). A role for these cells in T4's motion computation has been confirmed: Mi1 block flies are motion blind to ON-edge stimuli, while Tm3 input is specifically required to detect fast ON-edge motion in the preferred direction (*Ammer et al., 2015*). The polarities of transmission at both Mi1 and Tm3 inputs are likely to be the same because both appear to use acetylcholine as a neurotransmitter. In fact, a recent calcium imaging study has shown an enhanced T4 response when both cells are activated (*Strother et al., 2017*). Thus Mi1 and Tm3 could function as two excitatory input channels in an HR EMD circuit (*Figure 1A*), potentially depending on visual stimulus conditions such as different visual contrasts or velocities (*Ammer et al., 2015*).

This interpretation, however, runs counter to two features. First, assuming the measured directions of receptive field offsets between Mi1 and Tm3 (*Takemura et al., 2013*), the relative timing of the two inputs recorded by *Behnia et al. (2014)* is contrary to that in an HR EMD. Moreover, the magnitude of the spatial offset itself appears to be very small (*Takemura et al., 2013*; *Borst and Helmstaedter, 2015*). While our new data do not deny a role for Mi1 and Tm3 cells in motion detection, uncertainties in the direction of their small anatomical offsets (see *Figure 4—figure*

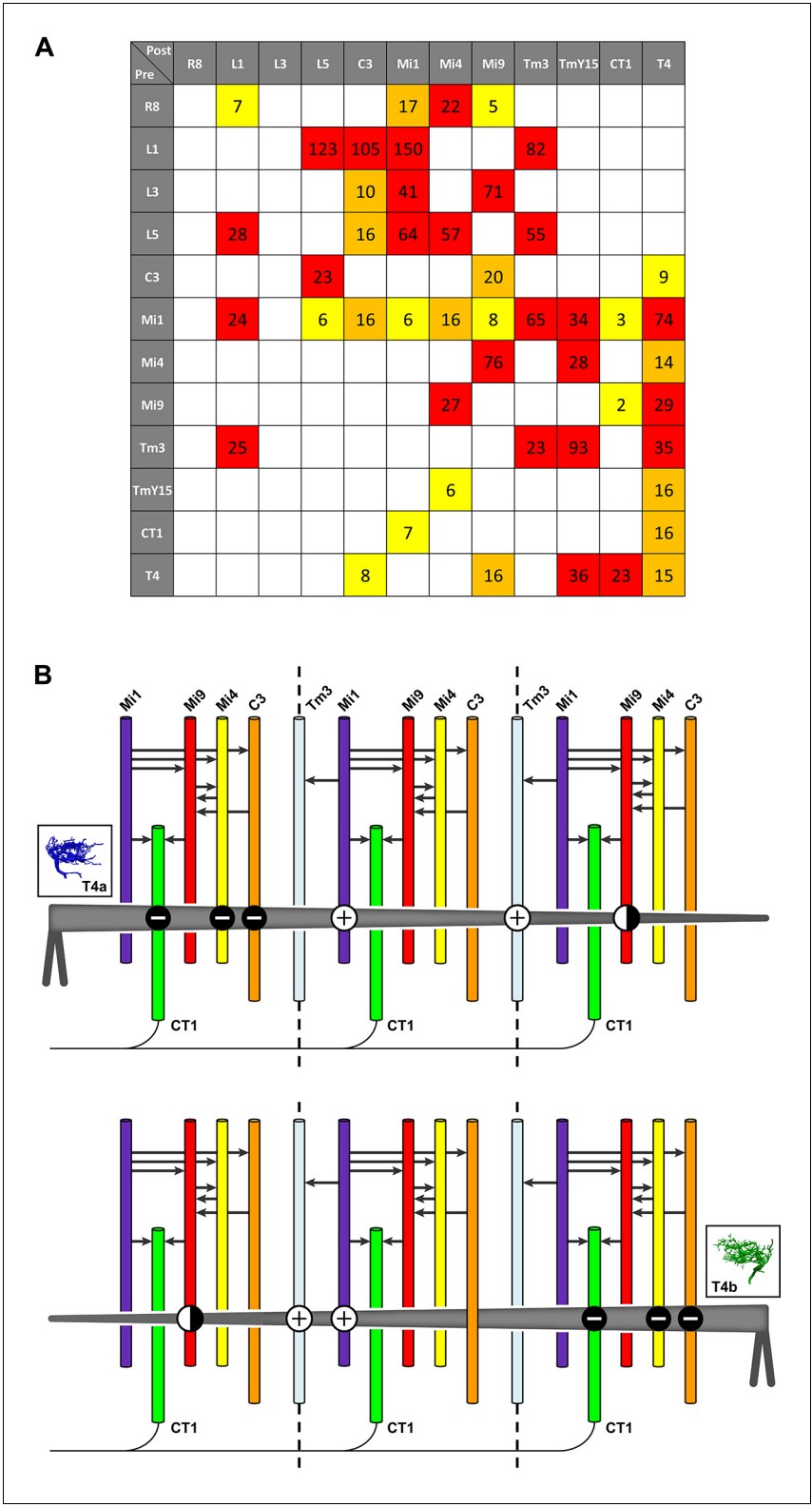

**Figure 6.** Pathways to T4 input channels. (**A**) Matrix of connections between R8, L1, L3, L5, C3, Mi1, Mi4, Mi9, Tm3, TmY15, CT1, and T4. Pathways are colour-coded at intercepts by strength, to highlight those with <10 synapses (yellow), 10–20 synapses (orange) and with >20 synapses (red). Note that single Tm3, TmY15 and T4 cells receive inputs from multiple cells of the same type because of their multicolumnar branching arbours. (**B**) T4 motion detection input circuits. T4a and T4b are shown, with opposite direction preferences. Each T4 extends its dendritic

*Figure 6 continued on next page*

*Figure 6 continued*

arbour across roughly three columns (dotted lines at column boundaries). Synaptic inputs from Mi1 and Tm3 spread along the shaft of T4 dendrites, whereas those from C3, CT1 and Mi4 are localized to T4's dendrite bases. Mi9 is located at the dendrite tips. Putative neurotransmitter phenotypes (see text and *Figure 5*) suggest that excitatory (+) and inhibitory (-) inputs onto T4's dendrites are segregated. These synaptic arrangements correspond not only to those predicted for a Hassenstein-Reichardt detector, as previously recognized, but also a Barlow-Levick-type detector (see text for details). Note that Mi1 provides inputs to all the presynaptic cells as well as directly to T4 cells (black arrows, see also matrix in (**A**)). Mi9's synaptic polarity (+ or -) is tentative (half-filled circles).

*supplement 2*) do make it difficult to uphold our previous conclusion that these offsets provide the core of an EMD.

Two additional input cell types, Mi9 and T4-T4 connections, that make synapses onto the leading edges of T4's dendrites (*Figure 4C and E*) could also function as excitatory input channels of an HR model. In this case they would work together with Mi1 and/or Tm3, qualified by their substantial spatial offsets (*Figure 4B*). However, possible roles of Mi9 depend on two properties of this cell type: the sign of the glutamatergic synapses and the nature of its light response – Mi9 might respond to OFF rather than ON-stimuli, with main synaptic input from L3 and Mi4 (*Figure 6A*). Indeed recent imaging results indicate that Mi9 does respond to OFF stimuli and is likely to provide inhibitory input to T4 dendrites (*Strother et al., 2017*; *Arenz et al., 2017*). T4-T4 synapses are also qualified to contribute to an HR mechanism: input from other T4s is spatially offset against the cells' preferred direction (*Figure 4E*) and predicted to be delayed since an additional synapse is involved (e.g. Mi1→T4→T4 vs Mi1→T4). However, blocking T4's synaptic outputs does not abolish directionally selective responses in T4 (*Haag et al., 2016*), suggesting that a mechanism based on T4-T4 synapses would, at least under some conditions, have to act in parallel with others. Such parallel contributions of several T4 input pathways could, for example, serve to broaden the T4 response curve for: stimulus velocity, which peaks at a temporal frequency of ~1 Hz (*Maisak et al., 2013*); or the range of stimulus modalities, for example to different light intensities or stimuli of differing spectral composition.

## Conclusion

In summary, we report the complete set of input cell types to direction-selective T4 cells, and reveal that these circuits are anatomically qualified to implement direction selectivity using both HR model preferred-direction enhancement and BL model null-direction suppression, as has indeed recently been reported from imaging studies in both T4 (*Haag et al., 2016*) and T5 cells (*Leong et al., 2016*). While not precluding the contribution of specific inputs to directional selectivity, synaptic inhibition from specific medulla cells could also be required for other reasons, and could function to signal something entirely different. For example, in addition to acting locally, CT1 is anatomically qualified to modulate the gain of T4 and T5 neurons in response to overall levels of motion. Whatever mechanism(s) they support, our EM findings will be essential to understanding the detection of motion, in combination with specific functional tests.

## Materials and methods

### Tissue preparation and EM imaging

The heads of wild-type Canton-S female flies between 5 and 6 days post-eclosion were fixed and processed for EM by high-pressure freezing/freeze substitution, according to previously reported methods (*Takemura et al., 2013*). The brain tissue was embedded in Durcupan epoxy resin (Fluka) after fixation. The left part of the brain was used for imaging.

A block face series of 32,000 images was acquired by a Zeiss NVision FIBSEM (*Knott et al., 2008*) at a resolution of 10 nm per pixel. A focussed ion-beam was used to remove 2.5 nm of material from the sample block face. The images were aligned using affine transforms and consecutive sets of four images were summed to generate isotropic 10 nm voxels.

## Connectome reconstruction

To obtain a dense reconstruction of seven medulla columns, we used a sequence of automated image processing followed by manual proofreading with Raveler (https://openwiki.janelia.org/wiki/display/flyem/Raveler). First, the medulla region of interest (ROI) was identified, which contained seven medulla columns. The image set of the ROI was then divided into 234 smaller overlapping cubes, each $5 \times 5 \times 5$ µm (for a total reconstructed volume of ~30,000 µm$^3$). In each cube, presynaptic T-bar ribbons were first detected automatically (*Plaza et al., 2014*) and then annotated by hand with Raveler, along with corresponding postsynaptic sites. Manual annotation took a total of ~6000 person-hours to identify both pre- and postsynaptic sites, with 8–10 proofreaders working in parallel on different non-overlapping subsets of cubes. Next, a watershed algorithm was applied to the pixel-wise predictions to generate the initial over-segmentation of the volume. The over-fragmented volume was then refined using a context-aware two-stage agglomeration framework (*Parag et al., 2015*) to produce the final segmentation. The segmentation was improved through a manual process of 'focussed' proofreading in which we concentrated on the divisions between cells that are both biologically improbable and that produce important 3D neuron shapes, especially if these be doubtful (*Plaza et al., 2012*). An additional inspection centred on features thought improbable, such as 'orphan' fragments that failed to touch the surface of any cube. This manual proofreading took a total of ~4000 person-hours, with 8–10 proofreaders working simultaneously on different non-overlapping subsets of cubes. The cubes were then stitched together using a combination of automatic and manual operations. Neuron 3D reconstructions were generated with software NeuTu-EM (https://github.com/janelia-flyem/NeuTu/tree/flyem_release [*Zhao et al., 2017*], with a copy archived at https://github.com/elifesciences-publications/NeuTu), in which we examined synaptic locations and distributions, and T4's dendrite orientations. Neuron reconstructions were closely examined for proofreading errors. Cell types of reconstructed neurons were identified by comparing the shapes of their arbours with the shapes of previously reported neuron reconstructions and from light microscopy. Connectivity matrices were generated by combining the results of neuron reconstructions with those for synapse mappings. The plots of spatial offsets for the columnar inputs (Mi4, Mi9, C3, and CT1) were calculated as the centre of mass of locations weighted by synapse count of the associated column in an ideal hexagonal array, referenced to the centre of the same calculation for Mi1.

## Genetics, immunohistochemistry and light microscopy

Gal4 or Split-Gal4 (*Luan et al., 2006*; *Pfeiffer et al., 2010*) driver lines used to visualize T4 input neurons were, SS02432 (R48A07AD, VT046779DBD) (Mi9), SS01019 (R48A07AD; R13F11DBD) (Mi4), SS00809 (R19F01AD; R71D01DBD) (Mi1) (*Strother et al., 2017*), R26H09AD; R29G11DBD (C3) (*Tuthill et al., 2013*), SS01001 (R65E11AD; R20C09DBD) (CT1) and VT048653 (TmY15). The CT1 split-Gal4 line was constructed using hemidrivers selected on the basis of GAL4 line expression data (*Jenett et al., 2012*). VT048653, from the Vienna Tiles collection (*Kvon et al., 2014*), predominantly labels TmY15 cells in the medulla based on stochastic labelling data (41/63 MCFO-labeled medulla cells in 17 brains were TmY15 neurons) and the overall expression pattern of this driver line. For Multicolor Flp-out (MCFO) labelling (*Figure 2B*; *Figure 2—figure supplement 1A, B and D*), driver lines were crossed to MCFO-1 or MCFO-7 and processed for immunolabelling, mounted in DPX and imaged as described (*Nern et al., 2015*). Imaging and processing of brains with (split-)GAL4 driven expression of presynaptic and membrane markers (*Figure 2C and D*; *Figure 2—figure supplement 1C and E*) using pJFRC51-3XUAS-IVS-Syt::smHA in su(Hw)attP1 (*Nern et al., 2015*) and pJFRC225-5XUAS-IVS-myr::smFLAG in VK00005 (*Nern et al., 2015*) was as described (*Aso et al., 2014*). pJFRC12-10XUAS-IVS-myr::GFP in attP2 (*Pfeiffer et al., 2010*) was used for the neurotransmitter labelling experiments and the light microscopy data in *Figure 4—figure supplement 1*. Female flies were used for all experiments.

Light microscopy panels (*Figures 2B, C and D*; *Figure 2—figure supplement 1*) show reconstructed views that were generated using the neuronannotator mode of Vaa3D (*Peng et al., 2010*) and exported as TIFF format screenshots. Some panels (CT1 in *Figure 2C and D*; *Figure 2—figure supplement 1*, and TmY15 in *Figure 2B* and *Figure 2—figure supplement 1*) show different views of the same cells generated from the same confocal stack to illustrate different aspects of the

anatomy. Additional images (not shown) of other cells of these types showed the same general anatomical features.

Neurotransmitter immunolabellings used previously described antibodies against GAD1 (*Featherstone et al., 2000*) (a gift from F.R. Jackson obtained via Y. Aso), ChAT (mAb ChAT4B1) (*Takagawa and Salvaterra, 1996*) obtained from the Developmental Studies Hybridoma Bank and VGlut (*Daniels, 2004*, a gift from A. DiAntonio). For these experiments, fly brains were dissected in PBS, fixed in 2% (v/v) PFA (prepared from a 20% stock solution, Electron Microscopy Sciences, #15713) in PBS for 1 hr, washed with PBT (0.5% v/v TX-100 in PBS), incubated with PBT-NGS (PBT with 5% v/v Normal Goat Serum) for 30 min and then incubated for ~2 days with primary antibodies (anti-GAD1 1:1000 dilution or anti-VGlut 1:5000 and anti-ChAT 1:50) in PBT-NGS on a nutator at 4°C, followed by further washes with PBT and incubation with secondary antibodies (DyLight 649 AffiniPure Donkey anti Rabbit IgG (H+L), Jackson ImmunoResearch Laboratories, Inc., #711-495-152,1:300 dilution and DyLight 549 AffiniPure Donkey anti Mouse IgG (H+L), Jackson ImmunoResearch Laboratories, Inc #715-505-151, 1:300 dilution) in PBT-NGS for ~2 days. After additional washes, brains were mounted in SlowFade Gold antifade (ThermoFisher Scientific) under a no. 1.5 coverslip using small amounts of modeling clay as flexible spacers. Brains were imaged with the anterior brain surface towards the coverslip, except C3 cells, which were imaged from the posterior to favour their different cell body locations. Native fluorescence was used to detect the GFP marker in these experiments. Images were acquired on an LSM 710 confocal microscope with a 63 × 1.4 NA objective. With the exception of the anti-GAD1 labeling of Mi9 and Tm3, all images in *Figure 5* with labeling of a given neurotransmitter marker show brains processed together in the same tube. However, the labeled brains were mounted and imaged individually with some specimen specific adjustments of laser power and gain, and/or of brightness and contrast post-imaging. Multiple specimens were examined for each cell type with similar results. For CT1, 12/12 cells from six brains showed GAD1 immunoreactivity.

## Data access

Our 7-column medulla raw data have been published on the Janelia/Howard Hughes Medical Institute website, http://emdata.janelia.org/#/repo/medulla7column.

## Acknowledgements

We acknowledge the contributions and support of all members of the Janelia FlyEM project and proofreaders in the Meinertzhagen lab, as well as members of the Janelia FlyLight project for assistance in generating some of the light microscopy data. We thank Michael Reiser, Stephen Plaza and Kazunori Shinomiya for fruitful discussions and critical reading of the manuscript. We are grateful to James Strother, Fred Davis, Sean Eddy, Lee Henry, Allan Wong, and members of the Reiser lab for discussions and access to their data prior to publication. This study was funded and supported by the Howard Hughes Medical Institute and grant DIS-0000065 from NSERC.

## Additional information

### Funding

| Funder | Grant reference number | Author |
| --- | --- | --- |
| Howard Hughes Medical Institute | | Shin-ya Takemura<br>Aljoscha Nern<br>Dmitri B Chklovskii<br>Louis K Scheffer<br>Gerald M Rubin<br>Ian A Meinertzhagen |
| Natural Sciences and Engineering Research Council of Canada | DIS-0000065 | Ian A Meinertzhagen |

The funders had no role in study design, data collection and interpretation, or the decision to submit the work for publication.

## Author contributions

S-yT, Conceptualization, Formal analysis, Investigation, Writing—original draft, Writing—review and editing; AN, Investigation, Writing—review and editing; DBC, GMR, Conceptualization, Writing—review and editing; LKS, Conceptualization, Formal analysis, Writing—review and editing; IAM, Conceptualization, Writing—original draft, Writing—review and editing

## Author ORCIDs

Shin-ya Takemura, http://orcid.org/0000-0003-2400-6426
Aljoscha Nern, http://orcid.org/0000-0002-3822-489X
Louis K Scheffer, http://orcid.org/0000-0002-3289-6564
Gerald M Rubin, http://orcid.org/0000-0001-8762-8703
Ian A Meinertzhagen, http://orcid.org/0000-0002-6578-4526

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
