## [Decision Letter]

Thank you for submitting your article "The comprehensive connectome of a neural substrate for 'ON' motion detection in *Drosophila*" for consideration by *eLife*. Your article has been reviewed by three peer reviewers, one of whom, Alexander Borst (Reviewer #1), is a member of our Board of Reviewing Editors and the evaluation has been overseen by K VijayRaghavan as the Senior Editor. The following individual involved in review of your submission have agreed to reveal their identity: Karl Fischbach (Reviewer #3).

The reviewers have discussed the reviews with one another and the Reviewing Editor has drafted this decision to help you prepare a revised submission.

Summary:

There are two principles to extract the direction of motion: enhancement of excitation in the preferred direction (Hassenstein & Reichardt model, HR) and suppression of excitation in the null direction (Barlow-Levick model, BL). Recent papers have shown that both movement detection mechanisms (HR & BL) do not exclude each other but work together to shape motion sensitivity in T4 and T5 cells. T4 cells inside the 10th layer of the fly's medulla and the sibling T5 cells in the 1st lobula layer are known to be the most peripheral directionally sensitive columnar neurons in the optic lobe of *Drosophila*. There exist 4 T4 and T5 subtypes (a-d) per column, one for forward, one for backward, one for up, and one for down movement. T4 and T5 are representatives of two separate motion sensitive pathways, the L1-ON pathway and the L2-Off pathway respectively.

Takemura et al. use FIBSEM to reconstruct the synaptic inputs to direction selective neurons in the fly visual system. They provide what appears to be a fairly comprehensive accounting of the numbers and spatial distributions of synapses from eight presynaptic cell types onto 4 different T4 subtypes. In particular, the authors identify many more synapses compared to their previous EM study of this circuit, demonstrating that the superior resolution of FIBSEM was essential to accurately quantifying synaptic connectivity. While some input neurons like Mi1 and Tm3 make contact to the central location of the dendrite, irrespective of the preferred direction of the T4 cell, other input neurons switch their location on the T4-cell dendrite depending on the directional tuning of the specific T4-subtype, e.g. Mi9 and other T4 cells on the 'null-side' and Mi4, C3 and CT1 on the 'preferred-side'. In summary, the work represents an outstanding and important contribution which will form the basis of many forthcoming functional studies to unravel the cellular mechanisms of motion vision in the fly.

Essential revisions:

Here are the following major concerns:

1) The biggest problem is how to deal with this work's predecessor, Takemura et al., 2013, where the same authors had found a spatial offset between Mi1 and Tm3 that goes along with the preferred direction of the T4-subtype, and consequently had stated: 'We identified Mi1 and Tm3 inputs to T4 neurons as the two arms of a candidate correlation-based motion detector.' The new data set is unable to confirm this previous finding but brings up new cell types that clearly switch their contact site with the directional tuning of T4-subtypes. These new cell types had been a) missed or underestimated with respect to their number of contacts before, and b) not seen to switch position on the dendrite in a directional selective way. There appear to be two major reasons why the previous report was incomplete and misleading: first, the anisotropic resolution with 50 nm along the Z-axis only allowed about 50% of the T4 input synapses to be identified, and second, the 2013 paper densely reconstructed only a single column. This latter fact comes as a surprise because the previous report had stated: 'We therefore decided to reconstruct all the synaptic connections among neurons within a single reference column, as well as all the connections between the reference column and neurons within six nearest-neighbour columns (Figure 1).' This point weighs much more severely, because it introduced a systematic error, leaving out those cells that contact T4-cells in the neighboring columns and provide the spatial offset needed for the computation of motion direction. The current paper, thus, contains two major revisions: a) the previous offset between Mi1 and Tm3 cannot be confirmed, b) new cell types are found that show the required spatial offset. While the current paper addresses these critical points at several locations and qualifies the previous data accordingly, they don't do so consequently and are trying to rescue the previous hypothesis. The authors should make these statements most visibly and clear-cut in the Abstract, e.g. by saying: 'Consistent with our previous study, Mi1 and Tm3 form the two most frequent input synapses on T4 cells. We are, however, unable to reproduce the spatial offset between these cells reported earlier which lead to the proposal that they form the two arms of the motion detector. ' This is the cleanest way to deal with the problem. It is especially important since the previous paper caused strong resonance within and outside the *Drosophila* community, was heavily cited (more than 160 within the 3 years after its appearance), and served as the basis for several subsequent studies.

2) Related to this point, I don't understand, in principle, how a given offset between the receptive fields of Mi1 and Tm3 can give rise to direction selectivity to all four T4 cells within this column: if the offset is along the PD for the T4a cell, it would be in the ND for the T4b cell. The only way this can be done is by differently weighing the synaptic input from Mi1 and Tm3 across the dendrite of T4a versus T4b cells, irrespective of any potential offset between the receptive fields of Mi1 and Tm3. The data of respective synaptic input sites on the T4 cell dendrites have been collected and reveal no offset that goes along with the direction selectivity of the T4 cells. Thus, the previous statement cannot only be not confirmed but also be firmly refuted.

3) The data about Tm3 and TmY15 belong to the main Figure 3, and not to the supplement. Then, the same scale should be used in Figure 3 for all the different cell types. Also, since Figure 3 represent the main findings of this study in a nice and comprehensive way, they should be given more space.

4) The immuno-stainings in Figure 4—figure supplement 1 are incomplete with respect to (1) cell types (only 5 cell types are shown!), (2) transmitters (only GABAergic and cholinergic neurons are tested), as well as (3) with respect to the various proteins indicative for the respective transmitter (for GABA-ergic neurons, one should stain with abs against the GABA, GAD and the VGAT, for cholinergic neurons, one should also stain with abs against VAChT, etc pp.). This paper is about connectivity, and the treatment of the transmitter types in this supplement is rather cursory, compared to the analysis of the connectivity. The authors should at least try to provide immuno-stainings against ChAT and GAD1 for all input cells.

5) The finding that the anatomy allows for implementation of BL as well as of HR functionality, is not surprising as both functions have been shown to exist in parallel in functional studies. This is mentioned by the authors, but I suggest that they shape their discussion starting from that point and ask which components could fulfill which task. To me the discussion still reads as if the authors are looking for a decision which model holds true.

---

## [Author Response]

*Essential revisions:*

*Here are the following major concerns:*

*1) The biggest problem is how to deal with this work's predecessor, Takemura et al., 2013, where the same authors had found a spatial offset between Mi1 and Tm3 that goes along with the preferred direction of the T4-subtype, and consequently had stated: 'We identified Mi1 and Tm3 inputs to T4 neurons as the two arms of a candidate correlation-based motion detector.' The new data set is unable to confirm this previous finding but brings up new cell types that clearly switch their contact site with the directional tuning of T4-subtypes. These new cell types had been a) missed or underestimated with respect to their number of contacts before, and b) not seen to switch position on the dendrite in a directional selective way. There appear to be two major reasons why the previous report was incomplete and misleading: first, the anisotropic resolution with 50 nm along the Z-axis only allowed about 50% of the T4 input synapses to be identified, and second, the 2013 paper densely reconstructed only a single column. This latter fact comes as a surprise because the previous report had stated: 'We therefore decided to reconstruct all the synaptic connections among neurons within a single reference column, as well as all the connections between the reference column and neurons within six nearest-neighbour columns (Figure 1).' This point weighs much more severely, because it introduced a systematic error, leaving out those cells that contact T4-cells in the neighboring columns and provide the spatial offset needed for the computation of motion direction. The current paper, thus, contains two major revisions: a) the previous offset between Mi1 and Tm3 cannot be confirmed, b) new cell types are found that show the required spatial offset. While the current paper addresses these critical points at several locations and qualifies the previous data accordingly, they don't do so consequently and are trying to rescue the previous hypothesis. The authors should make these statements most visibly and clear-cut in the Abstract, e.g. by saying: 'Consistent with our previous study, Mi1 and Tm3 form the two most frequent input synapses on T4 cells. We are, however, unable to reproduce the spatial offset between these cells reported earlier which lead to the proposal that they form the two arms of the motion detector. ' This is the cleanest way to deal with the problem. It is especially important since the previous paper caused strong resonance within and outside the Drosophila community, was heavily cited (more than 160 within the 3 years after its appearance), and served as the basis for several subsequent studies.*

The reviewer is correct that the present paper failed to confirm the spatial offset between the receptive subfields of Mi1 and Tm3 that we had reported earlier in our 2013 paper. We recognize and have tried to deal with this point carefully in the main text as well as several supplemental figures. We agree with the reviewer that the Abstract should also mention this point. Following the reviewer’s suggestion, we therefore revised the Abstract as follows:

“Analysing computations in neural circuits often uses simplified models because the actual neuronal implementation is not known. For example, a problem in vision, how the eye detects image motion, has long been analysed using Hassenstein-Reichardt (HR) detector or Barlow-Levick (BL) models.[…] Our comprehensive connectome reveals complex circuits that include candidate anatomical substrates for both HR and BL types of motion detectors.”

*2) Related to this point, I don't understand, in principle, how a given offset between the receptive fields of Mi1 and Tm3 can give rise to direction selectivity to all four T4 cells within this column: if the offset is along the PD for the T4a cell, it would be in the ND for the T4b cell. The only way this can be done is by differently weighing the synaptic input from Mi1 and Tm3 across the dendrite of T4a versus T4b cells, irrespective of any potential offset between the receptive fields of Mi1 and Tm3. The data of respective synaptic input sites on the T4 cell dendrites have been collected and reveal no offset that goes along with the direction selectivity of the T4 cells. Thus, the previous statement cannot only be not confirmed but also be firmly refuted.*

We completely understand the reviewer’s concern, and the reviewer is correct that our previous statement is not confirmed by the current study. The current FIBSEM dataset has allowed us to trace synaptic connections in a 7-column volume with unprecedented accuracy. However, because of the different methods used, it was not possible to trace outside the etched volume using FIBSEM as we were able to do for the previous ssEM analysis, in which we could sparsely trace additional neurites beyond a single column (Figure 3—figure supplement 3A). Because of this limitation, we have no access to synapses from L1 to Tm3 that occur in the distal medulla outside the FIB-imaged volume. Unlike Mi1, Tm3 cells are multicolumnar and thus receive synaptic inputs from L1s that belong to columns lying beyond even the 7-column volume. Insofar as there is no obvious other way to assign Tm3 cells to specific columns by anatomical means (Figure 3—figure supplement 2), these synapses are required because calculating the preciseTm3-mediated receptive subfield for each T4 subtype is the only way to determine offsets between Tm3 and other cell types. This is different for unicolumnar neurons such as Mi1, which can be matched to a column unambiguously even without such detailed calculation. Despite this fact, we attempted to see the spatial offset between Mi1 and Tm3 using all the available data, but we can neither confirm nor refute the displacements between Mi1 and Tm3 field centers (Figure 3—figure supplement 3B). So, regrettably we cannot make any conclusion whether the spatial displacement between the receptive fields of Mi1 and Tm3 exists or not. We tried to explain this point thoroughly in the text (subsection “Spatial offset between different inputs distributed over T4 dendrites”).

*3) The data about Tm3 and TmY15 belong to the main Figure 3, and not to the supplement. Then, the same scale should be used in Figure 3 for all the different cell types. Also, since Figure 3 represent the main findings of this study in a nice and comprehensive way, they should be given more space.*

We agree with the reviewer’s suggestion, and Figure 3 has been moved to Figure 3—figure supplement 1 so that two panels B and C could be expanded. The sizes of panels C-E have not been changed, to enable the synaptic puncta to continue to be visualized clearly.

For the reviewer’s first point, let us explain the reason why we have not included the synapse plots of Tm3 and TmY15. The five input cell types in Figure 3 (revised Figure 3) are all unicolumnar cells, and hence can be explicitly assigned to their associated columns. However, as also explained in our response to point 2 above, Tm3 and TmY15 are multicolumnar neurons that give no clue as to the exact column to which we should assign them. We therefore have no compatible way to plot their synaptic counts into the same hexagonal array (also because of the lack of access to the upstream input synapses as mentioned above). The plots of spatial offsets in Figure 3 were calculated based upon the weighted synapse counts of the associated columns in the hexagonal array. Therefore the receptive field centres of Tm3 and TmY15 cannot be calculated in the same way as for other input neurons. Although we lacked several of those significant synaptic inputs, we have tried to see if the spatial offset between Mi1 and Tm3 can be seen in the new reconstruction of FIBSEM according to the method we used previously (Takemura et al., 2013). We can, however, neither confirm nor refute the previously reported displacements between Mi1 and Tm3 field centres (Figure 3—figure supplement 3B).

*4) The immuno-stainings in Figure 4—figure supplement 1 are incomplete with respect to (1) cell types (only 5 cell types are shown!), (2) transmitters (only GABAergic and cholinergic neurons are tested), as well as (3) with respect to the various proteins indicative for the respective transmitter (for GABA-ergic neurons, one should stain with abs against the GABA, GAD and the VGAT, for cholinergic neurons, one should also stain with abs against VAChT, etc pp.). This paper is about connectivity, and the treatment of the transmitter types in this supplement is rather cursory, compared to the analysis of the connectivity. The authors should at least try to provide immuno-stainings against ChAT and GAD1 for all input cells.*

Following this suggestion, we have now provided the immunostaining results for two more cell types, Mi9 and Tm3 and also included immunolabelling with a marker for glutamatergic neurons (anti-VGlut).

*5) The finding that the anatomy allows for implementation of BL as well as of HR functionality, is not surprising as both functions have been shown to exist in parallel in functional studies. This is mentioned by the authors, but I suggest that they shape their discussion starting from that point and ask which components could fulfill which task. To me the discussion still reads as if the authors are looking for a decision which model holds true.*

At the reviewer’s suggestion, we revised the discussion part of the main text by adding some sentences mentioning the recent functional studies. Our data suggest that both models may indeed co-exist using different paired inputs. Here is a highlight from our revision:

“The finding of additional T4 inputs and their synaptic arrangements allows us to compare the predictions from functional studies with the requirements of different models of motion detection. Our findings reveal potential substrates for both BL- and HR-type motion detectors. A recent imaging study reported that T4 cells indeed use both HR model preferred-direction enhancement and BL model null-direction suppression (Haag et al., 2016). Which individual inputs onto T4’s dendrites fulfill the requirements for enhancement and suppression of directional selectivity? Of the synaptic inputs to T4 that are anatomically qualified to serve as two channels of a motion detector, several could support BL motion detectors (Figure 1). First, […]”